# Long-Term Treatment with Low-Level Arsenite Induces Aberrant Proliferation and Migration via Redox Rebalance in Human Urothelial Cells

**DOI:** 10.3390/cells14120912

**Published:** 2025-06-16

**Authors:** Xiangli Yan, Qing Zhou, Shuhua Xi, Peiyu Jin

**Affiliations:** 1Department of Nutrition and Food Hygiene, School of Public Health, China Medical University, No 77 Puhe Road, Shenyang North New Area, Shenyang 110122, China; 2023120320@cmu.edu.cn; 2Department of Occupational and Environmental Health, School of Public Health, China Medical University, Shenyang 110122, China; qzhou94@cmu.edu.cn (Q.Z.); shxi@cmu.edu.cn (S.X.); 3Key Laboratory of Environmental Stress and Chronic Disease Control & Prevention Ministry of Education, School of Public Health, China Medical University, Shenyang 110122, China

**Keywords:** arsenite, urothelial cells, redox balance, thioredoxin system, glutathione system

## Abstract

Chronic exposure to arsenic via drinking water can induce bladder cancer in humans. Nevertheless, there is little knowledge about the precise mechanisms of this. Abnormal elevations in cell proliferation and migration have repeatedly been identified as the first cellular traits of carcinogenesis. The aims of this study are to uncover the molecular mechanisms underlying arsenic-induced aberrant proliferation and migration of uroepithelium cells by exploring the role of cellular redox modulation. Our results show significant elevations in the levels of ROS and GSH, Trx1, components of the Nrf2 system, and NLRP3 inflammasome activity in the cells chronically treated with arsenite, which also experienced markedly enhanced proliferation and migration capacities. Additionally, ROS inhibitors, NLRP3, and the above antioxidant system could suppress this enhancement of the proliferation and migration capacities and reverse overexpression in these cells. However, only the AKT and ERK inhibitors were capable of reversing EGF, TGFα, and HSP90 overexpression. In conclusion, our findings indicate that the cellular redox status in the uroepithelium following chronic treatment with low-level arsenite was rebalanced due to ROS overproduction and compensatory upregulation of the redox control systems, which may allow ROS and Trx1 to be maintained at higher levels to facilitate cell proliferation and migration via overstimulation of the related signaling pathways.

## 1. Introduction

Epidemiological investigations have found that arsenic levels in drinking water are dose-dependently associated with the risk of developing bladder cancer in humans [1]. Findings from a two-year animal study showed that long-term exposure to dimethylarsinic acid (DMA) via drinking water could induce bladder carcinomas in male F344 rats [2]. We recently reported that exposure to arsenite or DMA for 12 weeks could result in simple hyperplasia of the bladder uroepithelium in female F344 rats [3]. Moreover, we also observed that prolonged exposure of the uroepithelium to 0.5 μM arsenite for 40 weeks could induce hyperactivation of EGF receptors and subsequent aberrant cell proliferation and migration, as well as malignant transformation [4]. Additionally, we found that the EGF, TGFα, and HSP90 protein levels may be elevated and involved in the hyperactivation of EGF receptors in these cells. Currently, there is compelling evidence for a relationship between arsenic exposure and bladder cancer, but the exact molecular pathways behind arsenic-stimulated tumorigenesis in the bladder have not been completely elucidated.

Among the factors involving tumorigenesis and cancer progression, oxidative stress is an important and well-researched factor. Oxidative stress arises from the imbalance between ROS production and scavenging by antioxidant systems, which is a well-established principal risk factor for cancer. Extensive investigations have disclosed that excess ROS can oxidize biological macromolecules, leading to tumorigenesis. Consequently, antioxidants can prevent tumorigenesis by scavenging the excessive ROS inside cells [5]. Substantial evidence indicates that ROS can act as signaling molecules and play key roles in the regulation of cellular signaling transduction, involving cell proliferation, differentiation, and survival [6]. In fact, compared to normal cells, cancer cells produce more ROS because of their higher metabolic rates. However, compensatory upregulation of antioxidant systems can offset the ROS overload inside cells. It has been well-documented that the intracellular redox status controls cell proliferation, migration, and apoptosis. There is a complex set of redox control systems for modulating the redox status in vivo in order to avoid redox imbalance. Of these, the GSH and Trx systems, as well as the Nrf2 pathway, represent the major intracellular antioxidant mechanisms. They complement each other in maintaining intracellular redox homeostasis; perturbing one is likely to affect the others [7,8]. There is recent evidence that disruption of the Trx or GSH systems may function in a wide range of conditions, including cancer [9,10,11].

The Trx system consists of Trx, TrxR, and an endogenous inhibitor of Trx, TXNIP. Trx has a conserved sequence of Cys–Gly–Pro–Cys in its redox catalytic site. Trx has two primary variants in mammalian cells: Trx1, which is present in the cytosol and nucleus, and Trx2, which is found in mitochondria. Trx1 has been more extensively studied and is believed to perform multiple functions, including antioxidant, growth stimulation, and apoptosis inhibition. Since modifying the redox status of proteins may alter the DNA binding ability of transcriptional factors and activity of enzymatic proteins, it is believed to be the main mechanism underlying Trx1-induced growth stimulation and is being increasingly linked to the development and expression of cancer phenotypes [12]. Mammalian TrxR is a selenoenzyme that catalyzes the reduction of the oxidized form of Trx.

Similar to the Trx system, the GSH system is also reported to contribute to cell differentiation, proliferation, and apoptosis. GSH is the most abundant antioxidant inside cells, and disruptions in the balance of GSH levels have a role in the development and progression of several human disorders, such as cancer [13]. Tumor cells derived from TrxR1−/− mice have been shown to retain the typical abilities to proliferate, form colonies, and develop into tumors. However, they are exceptionally susceptible to the impacts of GSH inhibitors [14]. Recent investigations have shown that the integration of GSH and glutaredoxin may effectively decrease Trx1 activity in vitro, particularly when TrxR1 is inhibited. In contrast, integrated suppression of GSH and Trx systems could cause overoxidation of Trx1 and reduce the viability of HeLa cells [15]. In addition, dual inhibition of the Trx and GSH systems could activate the Nrf2 pathway, suggesting that the Trx1 and GSH systems can be driven by this pathway [16]. By upregulating the gene expression of antioxidants, Nrf2 functions as the central player in the modulation of cellular redox homeostasis. Thus, upregulated antioxidant systems are important for maintaining ROS at a higher level inside cells to stimulate their fast metabolism and malignant proliferation, since they can resist the DNA damage and disturbance of protein homeostasis induced by excessive ROS and are protected from programmed cell death. Currently, there has been a growing appreciation that inhibition of antioxidant systems may represent an effective anticancer approach against tumors of multiple origins [17].

Trx1 levels increase in many human tumors, and this increase is closely associated with aggressive tumor growth. Thus, Trx1 has become a hot topic of study in the field of tumor molecular biology. A positive connection between serum Trx1 levels and the quantity of total water arsenic intake or urinary arsenic species has been observed in an epidemiological study [18]. Furthermore, either short- or long-term treatment with low-dose arsenite has been demonstrated to stimulate increased Trx1 expression in human GM847 fibroblasts [19]. The expression levels of Trx1 and TrxR1 were also observed to increase in a dose-dependent manner in arsenite-treated human liver HHL-5 cells and rat liver [20]. However, to the best of our knowledge, no studies have reported the role of the Trx system in the tumorigenesis of the urothelial cells following chronic treatment with arsenite.

In the present study, an in vitro experiment was performed using human urothelial cells to explore the roles of cellular redox modulation in aberrant cell proliferation and migration following extended exposure to arsenite, with the aim of uncovering the underlying molecular mechanisms. Findings from this work may promote further understanding of the precise mechanisms of urothelial tumorigenesis induced by long-term exposure to arsenic.

## 2. Materials and Methods

### 2.1. Antibodies and Different Reagents

The antibodies, reagents, and kits used in this study are as follows: ERK (Cell Signaling Technology, Danvers, MA, USA, 9102); p-ERK (Cell Signaling Technology, 9101); p38 (Cell Signaling Technology, 9212); p-p38(Cell Signaling Technology, 9211); JNK (Cell Signaling Technology, 9252); p-JNK (Cell Signaling Technology, 9251); AKT (Cell Signaling Technology, 9272); p-AKT (Cell Signaling Technology, 9271); Grx1 (ABclonal Technology, Wuhan, China, A23246); Trx1 (ABclonal Technology, A4024,); TrxR1 (ABclonal Technology, A4725); NLRP3 (ABclonal Technology, A24294,); TXNIP (ABclonal Technology, A9342); Caspase1 (ABclonal Technology, A0964); ASC (ABclonal Technology, A22046); IL-18 (ABclonal Technology, A23999); IL-1β (ABclonal Technology, A22257); P65 (Proteintech, Wuhan, China, 10745-1-AP); p-p65 (Proteintech, 82335-1-RR); EGF (ABclonal Technology, A26211); HSP90 (ABclonal Technology, A5027); TGFα (ABclonal Technology, A0337); Nrf2 (ABclonal Technology, A3577); β-actin (ABclonal Technology, AC038); U0126 (Selleck Chemicals, Houston, TX, USA, S1102); SB203580 (Selleck Chemicals, S1076); SP600125 (Selleck Chemicals, S1460), LY294002 (Selleck Chemicals, S1105); PDTC (Selleck Chemicals, S3633); BSO (Selleck Chemicals, S9728); PX-12 (MedChemExpress, Monmouth Junction, NJ, USA, HY-13734); auranofin (MedChemExpress, HY-B1123); MCC950 (MedChemExpress, HY-12815); melatonin (MedChemExpress, HY-B0075); TBHQ (MedChemExpress, HY-100489); ML385 (MedChemExpress, HY-100523). ELISA assay kits for EGF (YX-E10084), TGFα (YX-E10114), IL-18 (YX-E10092), and IL-1β (YX-EILB1) were purchased from the Baolai Biotechnology Company, Nanjing, China. The assay kit for intracellular ROS levels (KGA7308) was a product of Jiangsu Kaiji Biotechnology Co., Ltd., Nanjing, China. The assay kits for the GSH levels (BC1175) and activities of TrxR1 (BC1150) and GR (BC1160) were purchased from the Solarbio Science and Technology Co., Ltd., Beijing, China.

### 2.2. Cell Culture and Treatment

SV-HUC-1 cells were obtained from the Cell Bank and were acquired according to the methodology outlined in the study [21,22] and cultured continuously for 40 weeks with or without 0.5 µM arsenite treatment. When the cells reached 80% confluence, the cells were treated for 24 h with either 5 μM melatonin (antioxidant), 20 μM PX-12 (Trx1 inhibitor), 5 μM auranofin (TrxR1 inhibitor), 100 μM BSO (GSH synthetase inhibitor), 10 μM ML385 (Nrf2 inhibitor), 10 μM TBHQ (Nrf2 activator), 10 μM U0126 (ERK inhibitor), 10 μM SP600125 (p38 inhibitor), 10 μM SB203580 (JNK inhibitor), 10 μM LY294002 (PI3K/AKT inhibitor), 50 μM PDTC (NF-κB inhibitor), or 10 μM MCC950 (NLRP3 inhibitor), as indicated.

### 2.3. Cell Viability Assay

Cell proliferation was evaluated through the conversion of MTS to formazan, where mitochondrial activity is positively correlated with cell number. When the cells seeded in a 96-well culture plate had grown to 80% confluence, 20 µL MTS was added to the culture and incubated for 2 h at 37 °C. Thereafter, the optical density (OD) of each well was measured at 490 nm using a microplate reader. Three parallel samples were included in each group. The mean of the three samples in the control group was first determined and then used to correct the mean of the experimental group.

### 2.4. Assessment of Intracellular ROS Levels

Briefly, the cells were cultured in a 96-well plate and incubated with DCFH-DA for 30 min at 37 °C in the dark. The ROS levels in cells were evaluated according to the intensity of DCF fluorescence. The ROS levels were expressed as the OD ratio between the treatment and control groups. Three parallel samples were included in each group. The mean of the three samples in the control group was first determined and then used to correct the mean of the experimental group.

### 2.5. Evaluation of TrxR1 and GR Enzymatic Activity Together with Intracellular GSH Levels

Using specific assay kits and adhering to the manufacturer’s protocol, the activities of TrxR1 and GR, as well as cellular GSH levels, were assessed. Three parallel samples were included in each group. The mean of the three samples in the control group was first determined and then used to correct the mean of the experimental group.

### 2.6. Transwell Assay

The cells, which were in a state of suspension, were placed into the top chamber of a Transwell plate with serum-free media. The culture medium, which contained 10% FBS, was introduced into the bottom chamber as a chemoattractant. Afterward, the cells were cultured for 24 h at 37 °C. The cells that had migrated to the underside of the membrane were fixed by treating with a 4% paraformaldehyde solution and then stained with crystal violet. The stained cells were imaged using a microscope. Thereafter, the cells were destained and transferred into a 96-well plate for determining OD at 570 nm. The migrated cell number was computed and expressed as the OD ratio between the treatment and control groups.

### 2.7. Wound Healing Assay

The cell proliferation and migration activities were also evaluated using wound healing assays. Once the cells in a 6-well plate had reached 80% confluence, a “wound” was induced by scratching with a pipette tip. The wound width was determined using a photograph taken under a microscope at 0, 12, and 24 h after scratching. ImageJ software 1.53 was utilized to calculate the healing area.

### 2.8. Western Blot Analysis

In this study, we strictly followed the established procedures described in our previous paper [21,22]. An equal amount of protein (30 µg per lane) was separated via electrophoresis on a 10% SDS polyacrylamide gel and then transferred onto a PVDF membrane. Following transfer, the blots were incubated with blocking buffer and labeled with primary antibodies against EGF (1:200), TGFα (1:200), HSP90 (1:2000), p65 (1:500), AKT (1:500), JNK (1:500), ERK (1:500), p38 (1:500), p-p65 (1:500), p-AKT (1:500), p-JNK (1:500), p-ERK (1:500), p-p38 (1:500), Nrf2 (1:500), Grx1 (1:500), Trx1 (1:500), TXNIP (1:500), NLRP3 (1:1000), ASC (1:500), Caspase1 (1:1000), IL-18 (1:500), IL-1β (1:500), and β-actin (1:3000) with overnight incubation at 4 °C. Subsequently, the target bands were detected by incubation with horseradish peroxidase-conjugated secondary antibodies (1:5000) for 1 h at room temperature. Finally, the resulting infrared signal was visualized using an LI-COR Odyssey Infrared Imaging system. Quantification was performed through densitometry using ImageJ and normalized against β-actin from the same blot.

### 2.9. Animals and Treatments

Sixty female Fischer 344/DuCrj (F344) rats (80–90 g) were obtained at five weeks of age from the Beijing Vital River Laboratory Animal Technology Co. Ltd., China. The animals were selected because they are more sensitive to the carcinogenic effects of DMA^V^ on the urothelium. On arrival, the animals were kept in a ventilated room at a temperature of 23–27 °C, with a humidity of 55–60% and a 12-h light/dark cycle, and were group-housed (five per cage) in polycarbonate cages with wood chips for bedding. They had free access to a standard rodent diet and drinking water throughout the study. The rats were randomly divided into three groups of 20 rats each, using a weight stratification method. They received unaltered drinking water (control group), drinking water containing 50 ppm arsenic as 87 ppm sodium arsenite (arsenite-treated group), and drinking water containing 108 ppm arsenic as 200 ppm DMA^V^ (DMA^V^-treated group), respectively, for 12 weeks. Doses of arsenite and DMA^V^ used in this study were determined according to the results of prior studies.

### 2.10. ELISA Test

In the culture media, mature forms of IL-18, IL-1β, EGF, and TGFα were measured using specific ELISA kits following the manufacturer’s recommendations. In the rat blood and urine, mature forms of Trx1, TrxR1, GSH, GR, IL-1β, and IL-18 were measured using specific ELISA kits following the manufacturer’s recommendations. The concentrations are expressed as pg/mL.

### 2.11. Immunofluorescence Staining

The urothelium slides were routinely deparaffinized and rehydrated. Thereafter, the next steps are antigen retrieval. They were then incubated with primary antibodies against Trx1 (×100), TrxR1 (×100), TXNIP (×100), and NLRP3 (×100) in a humidity chamber at 4 °C overnight. Subsequently, the slides were incubated with secondary antibodies at 37 °C for 60 min. DAPI was applied to stain the cell nuclei at the final step. They were captured using a fluorescence microscope (Olympus BX-61, Tokyo, Japan).

### 2.12. Statistical Analysis

All experiments in this investigation were performed in at least triplicate. The data were normalized and are reported as the mean ± standard deviation (SD). The analysis was conducted with SPSS for Windows (version 17.0, SPSS Inc., Chicago, IL, USA). The presence of significant variations between two separate groups or between various groups was assessed using the Student’s unpaired *t*-test or analysis of variance test (one-way ANOVA), followed by the Student–Newman–Keuls (SNK) test. Statistical significance is defined as a *p*-value below 0.05.

## 3. Results

### 3.1. Alteration of Redox Control Systems and Their Roles in the Proliferation and Migration of SV-HUC-1 Cells Following Extended Arsenite Exposure

Here, we first examined ROS and GSH levels, levels of Trx1 and Nrf2 protein expression, as well as the activities of GR and TrxR1 in cells chronically treated with arsenite, and then melatonin, PX-12, auranofin, BSO, ML385, and TBHQ were, respectively, applied to explore their roles in arsenite-induced aberrant proliferation and migration.

We found that there were significantly elevated ROS and GSH levels, Trx1 and Nrf2 protein levels, as well as GR and TrxR1 activities in arsenite-treated cells (Figure 1A–D). These cells also exhibited a significant increase in their abilities to proliferate and migrate, as assessed via cell viability, Transwell, and wound healing assays (Figure 1E–G). In the uroepithelium, long-term exposure to low-dose arsenite may induce cell self-renewal, apoptosis resistance, energy metabolism reprogramming, enhanced glucose uptake, and aerobic glycolysis, which contributes to increases in cell viability and proliferation. In contrast, treatment with melatonin, PX-12, auranofin, BSO, TBHQ, and ML385 could suppress these enhancements in the cell proliferation and migration capacities, although melatonin and TBHQ could reduce the ROS levels in these cells, while PX-12, auranofin, BSO, and ML385 elevated them (Figure 1A,E–G, Figure 2A,E–G and Figure 3A,E–G). Moreover, melatonin, PX-12, auranofin, and TBHQ could enhance the GSH levels and GR activities in these cells, whereas BSO and ML385 reduced them (Figure 1B,D, Figure 2B,D and Figure 3B,D). Finally, melatonin, TBHQ, and BSO could also elevate the Trx1 protein levels and TrxR1 activities in these cells, whereas PX-12, auranofin, and ML385 conversely reduced them (Figure 1C,D, Figure 2C,D and Figure 3C,D).

Additionally, our findings from the animal experiment also revealed that the Trx1 and TrxR1 protein levels were significantly increased in the urothelium cells, blood, and urine of female F344 rats treated with arsenite or DMA via drinking water for 12 weeks (Appendix A). Furthermore, the GSH and GR levels in the urine of these rats were also significantly increased (shown in Appendix A).

Collectively, our data suggest that both the oxidative stress and antioxidant capacities in the uroepithelium were upregulated following long-term treatment with arsenite. The cellular redox status in these cells might rebalance due to ROS overproduction and compensatory upregulation of the redox control systems, which might contribute to the enhanced cell proliferation and migration capacities, as disruption of the cellular redox balance via suppressing or enhancing antioxidant systems could impair cell proliferation and migration capacities.

### 3.2. Mechanisms of EGF, TGFα, and HSP90 Overexpression, as Well as Aberrant Cell Proliferation and Migration, in SV-HUC-1 Cells Following Extended Arsenite Treatment

In this part, we first investigated the alteration in activation of the NLRP3 inflammasome and MAPK, PI3K/AKT, and NF-κB pathways in the cells chronically treated with arsenite. Thereafter, the effects of melatonin, PX-12, auranofin, BSO, ML385, and TBHQ on EGF, TGFα, and HSP90 overexpression, as well as the activation of MAPK, PI3K/AKT, and NF-κB pathways, were explored. Finally, the pathways involving EGF, TGFα, and HSP90 overexpression, as well as aberrant proliferation and migration, were uncovered by using U0126, SP600125, SB203580, LY294002, PDTC, and MCC950 in these cells.

Here, our results revealed that prolonged exposure to arsenite may significantly elevate the protein levels of EGF, TGFα, HSP90, NLRP3, ASC, Caspase1, and IL-18/-1β, as well as the phosphorylated levels of p65, AKT, JNK, ERK, and p38 in cells. Additionally, it can also raise the amounts of EGF, TGFα, and IL-18/-1β in the surrounding medium (Figure 4A–I). Moreover, treatment with melatonin, PX-12, auranofin, BSO, TBHQ, and ML385 could reduce the EGF, TGFα, and HSP90 protein levels and phosphorylated levels of p65, AKT, JNK, ERK, and p38 in cells, as well as levels of EGF and TGFα in their media (Figure 4A–I, Figure 5A–E and Figure 6A–E). Thus, our data indicate that ROS overproduction and compensatory upregulation of the redox control systems might contribute to the overexpression of EGF, TGFα, and HSP90, as well as activation of the ERK, JNK, p38 MAPK, PI3K/AKT, and NF-κB pathways, in uroepithelium chronically exposed to arsenite.

### 3.3. Effects of the NLRP3 Inflammasome on Cell Proliferation and Migration, Enhancement of HSP90, TGFα and EGF Levels, and Activation of Related Signaling Pathways Following Long-Term Arsenite Treatment

On the other hand, our results also revealed that treatment with melatonin could reduce the NLRP3, ASC, Caspase1, and IL-18/1β protein levels in cells and IL-18/1β levels in their media (Figure 4C–F). Moreover, MCC950, a specific inhibitor of NLRP3 activation, could suppress cell proliferation and migration capacities, overexpression of EGF and TGFα, as well as activation of ERK, JNK, p38 MAPK, PI3K/AKT, and NF-κB signaling pathways in these cells (Figure 7A–H). Additionally, the animal experiment outcomes revealed that NLRP3 protein levels in the urothelial cells and IL-18/-1β levels were also increased significantly in the blood and urine of female F344 rats treated with arsenite or DMA throughout drinking water for 12 weeks (Appendix A). It is notable that the TXNIP protein levels decreased significantly in the urothelial cells chronically exposed to arsenite both in vivo and in vitro, and melatonin levels could be elevated in vitro (Figure 4C and Appendix A).

### 3.4. Effects of JNK, p38, ERK, AKT, or NF-κB on Cell Proliferation and Migration and the Enhancement of HSP90, TGFα, and EGF Following Long-Term Arsenite Treatment

Furthermore, our data showed that the inhibitors of ERK, JNK, p38 MAPK, PI3K/AKT, and NF-κB could all suppress the enhanced cell proliferation and migration capacities, and only inhibitors of ERK, PI3K/AKT, and NF-κB reduced the overexpression of EGF, TGFα, and HSP90 in these cells (Figure 8A–D). In addition, the inhibitors of ERK, JNK, p38 MAPK, and PI3K/AKT pathways could all hinder NF-κB activation in these cells (Figure 8E).

Altogether, findings from the experiments indicate that ROS overproduction and compensatory upregulation of the redox control systems might contribute to the activation of the signaling pathways, aberrant cell proliferation and migration, and to EGF, TGFα, and HSP90 overexpression in the uroepithelium following chronic exposure to arsenite. Activation of the ERK, JNK, p38 MAPK, and PI3K/AKT signaling pathways could all facilitate cell proliferation and migration through activating NF-κB; however, in these cells, only the ERK and PI3K/AKT signaling pathways contributed to EGF, TGFα, and HSP90 overexpression via activating NF-κB. In addition, the signaling pathways mediated by NLRP3 inflammasome activation could also contribute to cell proliferation and migration, but they might only be involved in EGF and TGFα overexpression in these cells.

## 4. Discussion

Here, we first observed that in the uroepithelium, ROS and GSH levels and Trx1 and Nrf2 protein expression, as well as TrxR1 and GR activities, were markedly increased following chronic treatment with arsenite both in vivo and in vitro, whereas TXNIP protein levels decreased significantly. Next, we found that suppression of Trx1 or TrxR1 could elevate GSH levels and GR activities. Conversely, inhibition of GSH synthesis could elevate Trx1 levels and TrxR1 activities. However, inhibition of either the Trx system or the GSH system could elevate ROS levels in these cells. This indicates that two major antioxidant systems inside cells—GSH and Trx1—compensate for and complement one another in maintaining homeostasis, further demonstrating the functional redundancy of Trx1 and GSH systems. Hence, the regulation of antioxidant systems could represent a promising cancer therapy target. As clearly shown in our results, ROS production in the uroepithelium may increase following prolonged treatment with arsenite, and to cope with oxidative stress, the GSH and Trx systems, as well as the Nrf2 pathway, were upregulated to buffer the excessive ROS in these cells. Consequently, the intracellular redox status was rebalanced at a higher ROS level due to the enhanced antioxidant capacity inside cells. Since the antioxidants, as well as activators or inhibitors of antioxidant systems, could all impair the activities of cell proliferation and migration, it is reasonable to speculate that ROS overproduction and enhanced antioxidant capacity inside cells could all contribute to aberrant cell proliferation and migration in the uroepithelium following chronic treatment with arsenite. The present results also reveal that ROS overproduction might be involved in the downregulation of TXNIP in these cells, since antioxidants could elevate TXNIP levels.

An increasing amount of evidence has demonstrated that Trx1 and TrxR1 are usually overexpressed in many tumors and closely linked with cancer development and aggressive behavior [23,24,25]. Inhibition of Trx1 has been shown to impede the growth, migration, and invasion of cells corresponding to multiple cancer types [26,27,28]. Analogously, the TrxR1 inhibitor, auranofin, has been reported to hinder the growth, migration, and clonogenic activity of many types of tumor cells [29,30,31]. Conversely, TXNIP is underexpressed in most cancer cells, and there is convincing evidence showing its levels are inversely correlated with tumor progression [32,33,34]. Recent research has shown that the expression of TXNIP was reduced in human bladder cancer. Furthermore, one study found that the absence of TXNIP expression promotes the development of bladder cancer in a mouse model [35]. Additionally, TXNIP overexpression has been reported to suppress the proliferation of carcinoma cells by triggering apoptosis via activating the MAPK signaling pathway [36]. These findings indicate that TXNIP acts as a tumor suppressor in the urothelium, which corresponds with its reduced expression in various cancers. We plan to further investigate the mechanism of TXNIP changes in the uroepithelium when chronically treated with arsenite.

The mechanisms underlying aberrant cell proliferation and migration, and EGF, TGFα, and HSP90 overexpression in the uroepithelium following chronic treatment with arsenite, are complex and may involve a variety of signaling pathways. Our results revealed that both the MAPK and PI3K/AKT signaling pathways are involved in cell proliferation and migration, as well as EGF, TGFα, and HSP90 overexpression via activating NF-κB in the studied cells. Our data further demonstrated that excessive ROS and upregulated redox systems could directly or indirectly activate these signaling pathways, since antioxidants and activators or inhibitors of redox control systems could all affect their activation. Additionally, our data showed that the signaling pathways mediated by NLRP3 inflammasome also contributed to aberrant proliferation and migration in these cells.

It has been established that ROS are produced in excess during arsenic metabolism, which can activate the MAPK, PI3K/AKT, Nrf2, and NF-κB pathways, which are thought to possess key roles in carcinogenesis in urothelial cells [37,38,39]. Our current results are in line with previous findings from our laboratory and others [40,41,42]. Additionally, Trx1 is reported to promote colorectal cancer invasion and metastasis via activating the ERK1/2 signaling pathway [25]. Trx1 can also promote the growth of neuroblastoma by activating the PI3K/AKT signaling pathway [43]. Therefore, upregulated Trx and GSH systems could also have an essential contribution to activating the abovementioned signaling pathways. Increases in GSH and Trx1 can alleviate the damage caused by excessive ROS levels, and maintaining ROS levels within an appropriate range is crucial for promoting the excessive proliferation of arsenite-treated urothelial cells. On the other hand, the current study demonstrated that ROS overproduction and enhanced redox control systems could also directly or indirectly activate the signaling pathways, contributing to EGF, TGFα, and HSP90 overexpression in the uroepithelium following chronic treatment with arsenite. Of the signaling pathways mentioned above, only ERK and PI3K/AKT were confirmed to be involved in the upregulation of EGF, TGFα, and HSP90 via activating NF-κB, and the signaling pathways mediated by the NLRP3 inflammasome only contribute to the upregulation of EGF and TGFα.

NLRP3 is the most extensively researched member of the NLR family and has a crucial function in several inflammation-related disorders. As a priming signal, ROS are known to contribute to the activation of the NLRP3 inflammasome that comprises NLRP3, ASC, and pro-Caspase1. After activation, pro-Caspase1 in the NLRP3 inflammasome is cleaved, and IL-1β and IL-18 are then generated by active Caspase1. The pro-inflammatory cytokines generated by the NLRP3 inflammasome may facilitate cell growth, migration, differentiation, and invasion by activating the related signaling pathways [44]. Intriguingly, in our recently published paper, the NLRP3 inflammasome was shown to be activated by either excess ROS or upregulated TXNIP in the uroepithelium following treatment with arsenite for 24 h, since TXNIP is an upstream partner of NLRP3 and the formation of TXNIP-NLRP3 is necessary for assembling and activating the NLRP3 inflammasome [22]. Moreover, in our previous study, we also revealed that a stimulated Trx system does not play a direct role in activating signaling pathways that contribute to EGF, TGFα, and HSP90 overexpression in the cells treated with arsenite for 24 h, since inhibitors of both Trx1 and TrxR1 did not suppress, but instead promoted, their activation. It is notable that the regulation mechanisms underlying the expression of TXNIP, EGF, TGFα, and HSP90 in the uroepithelium differ markedly depending on whether arsenite treatment is chronic or acute. Our findings suggest that short-term treatment with arsenite might upregulate the expression of EGF, TGFα, and HSP90 proteins, most probably via redox imbalance. However, for long-term treatment with arsenite, they might be upregulated in the urothelial cells mainly via redox rebalance.

ROS overproduction can cause damage to biomolecules and organelles, leading to inflammation, which is the main mechanism underlying hyperplasia and the development of carcinogenesis, suggesting that the regulatory relationship between redox rebalance and the NLRP3 inflammasome needs further investigation. Our findings indicate that the change trends in TXNIP and the NLRP3 inflammasome are not identical in the uroepithelium when chronically treated with arsenite. We will further explore the relationship between redox rebalance and TXNIP or the NLRP3 inflammasome so as to provide experimental reference data for revealing the mechanism of arsenic-induced hyperplasia and carcinogenesis of urothelial cells.

## 5. Conclusions

Overall, our results support the conclusion that after long-term treatment with low-dose arsenite, the cellular redox status in the uroepithelium is rebalanced due to ROS overproduction and compensatory upregulation of the redox control systems, which may allow ROS and Trx1 to be maintained at a higher level to facilitate cell proliferation and migration via overstimulation of the related signaling pathways. This study is the first to describe the involvement of the cellular redox state in arsenic-induced urothelial cancer based on our current understanding.

## Figures and Tables

**Figure 1 cells-14-00912-f001:**
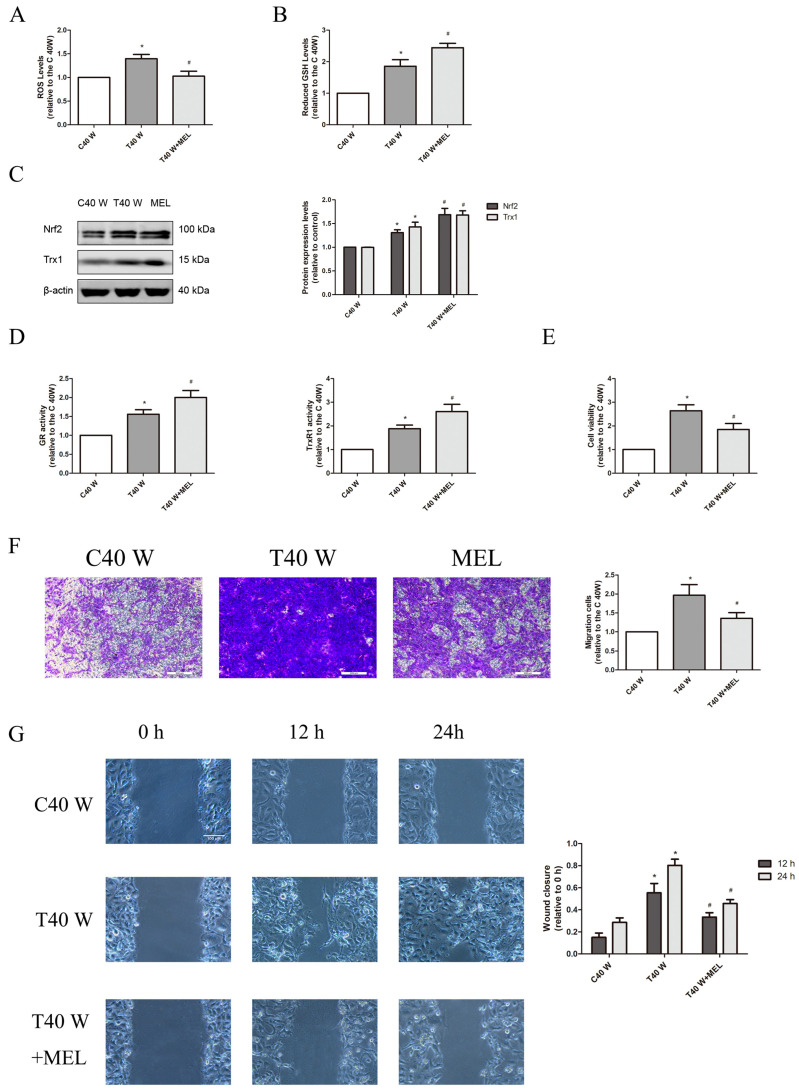
ROS are essential contributors to the proliferation and migration of cells following extended exposure to arsenite. (**A**) Evaluation of intracellular ROS levels. (**B**) Detection of intracellular GSH levels. (**C**) An analysis of Western blots and densitometric data was performed to observe effects on Nrf2 and Trx1 protein levels. (**D**) Activities assessment of GR and TrxR1. (**E**) The cell viability was ascertained with the MTS assay. (**F**) Transwell assays were implemented to estimate cell migration. Scale bar: 200 μM. (**G**) Cell migration capacities were assessed by wound-healing assays. Scale bar: 100 μM. The outcomes are reported as means ± SD, and a minimum of three separate experiments were conducted. * *p* < 0.05, vs. control group (untreated cells). # *p* < 0.05, vs. arsenite-treated group. C40 W, cells cultured for 40 weeks without arsenite treatment; T40 W, cells cultured for 40 weeks with arsenite treatment; MEL, melatonin. The scale bars in the photos represent 200 μM for transwell assay, and 100 μM for wound healing assay.

**Figure 2 cells-14-00912-f002:**
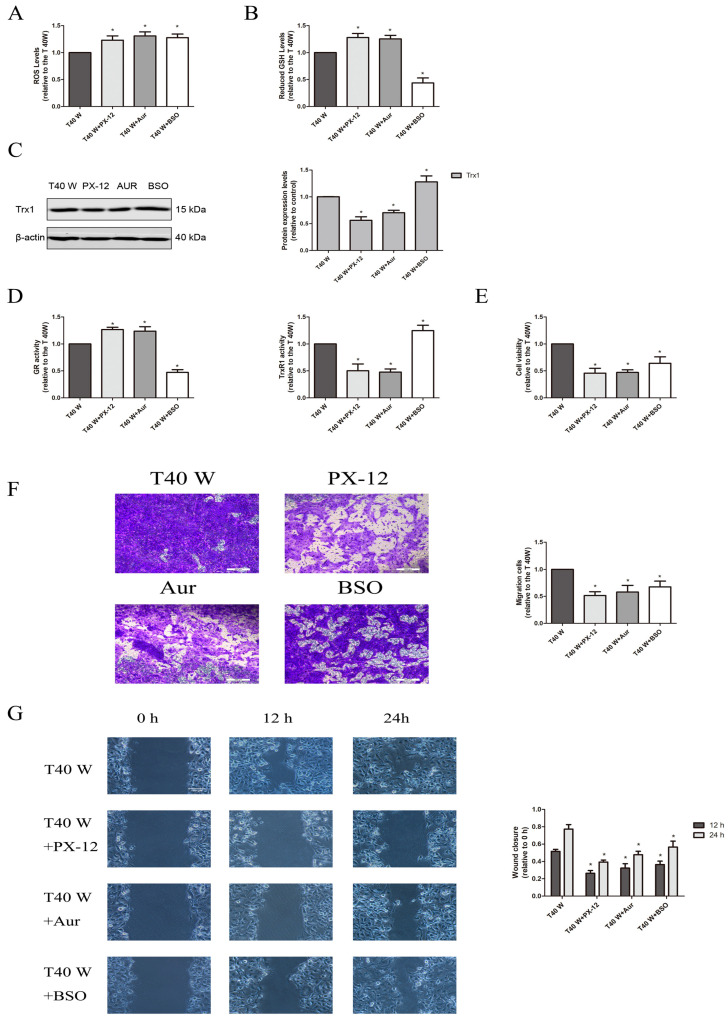
Trx1 and the GSH system contribute to cell proliferation and migration following long-term treatment with arsenite. Notes: PX-12, auranofin, or BSO was added to cells treated continuously with 0.5 µM arsenite for 40 weeks. (**A**) Evaluation of intracellular ROS levels. (**B**) Detection of intracellular GSH levels. (**C**) Analysis of Western blots and densitometric data showing the effects on Trx1 protein levels. (**D**) Activities assessment of GR and TrxR1. (**E**) Cell viability was ascertained according to the MTS assay. (**F**) Transwell assays for assessing cell migration. (**G**) Cell migration capacities were estimated using wound-healing assays. The outcomes are reported as means ± SD, and a minimum of three separate experiments were conducted. * *p* < 0.05, vs. arsenite-treated group. T40 W, cells cultured for 40 weeks with arsenite treatment; Aur, auranofin. The scale bars in the photos represent 200 μM for transwell assay, and 100 μM for wound healing assay.

**Figure 3 cells-14-00912-f003:**
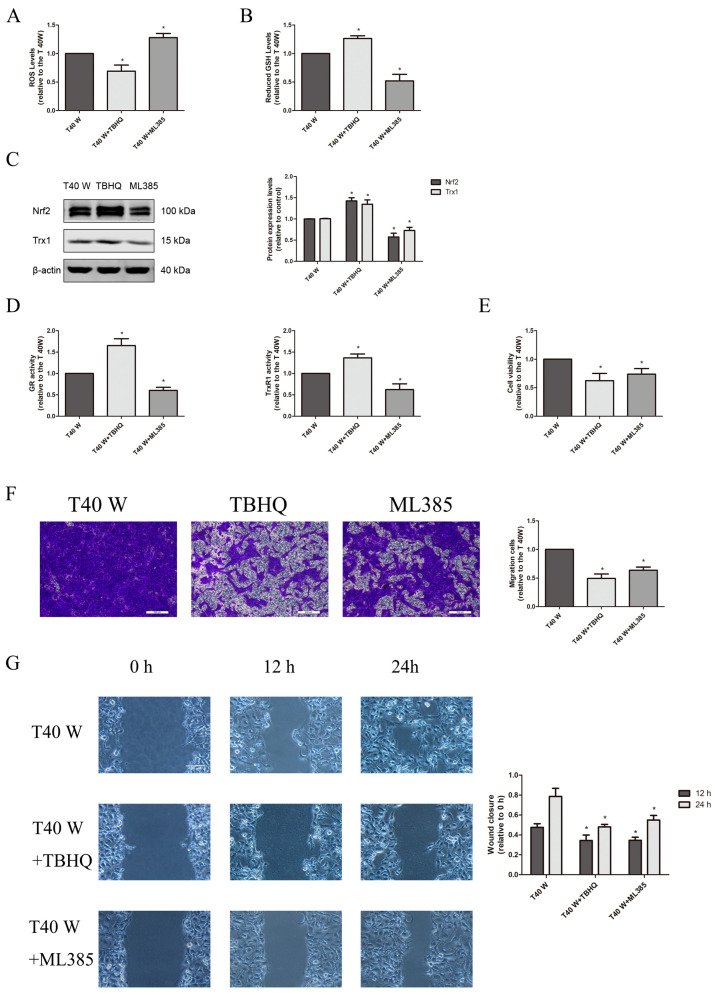
The involvement of the Nrf2 system in cell proliferation and migration is essential following extended exposure to arsenite. Either TBHQ or ML385 was added to cells treated continuously with 0.5 µM arsenite for 40 weeks. (**A**) Evaluation of intracellular ROS levels. (**B**) Detection of intracellular GSH levels. (**C**) Analysis of Western blots and densitometric data showing the effects on Nrf2 and Trx1 protein levels. (**D**) Assessment of GR and TrxR1 activities. (**E**) Cell viability was estimated according to the MTS assay. (**F**) Transwell assays for assessing cell migration. (**G**) Cell migration capacities were ascertained using wound-healing assays. The outcomes are reported as means ± SD, and a minimum of three separate experiments were conducted. * *p* < 0.05, vs. arsenite-treated group. T40 W, cells cultured for 40 weeks with arsenite treatment. The scale bars in the photos represent 200 μM for transwell assay, and 100 μM for wound healing assay.

**Figure 4 cells-14-00912-f004:**
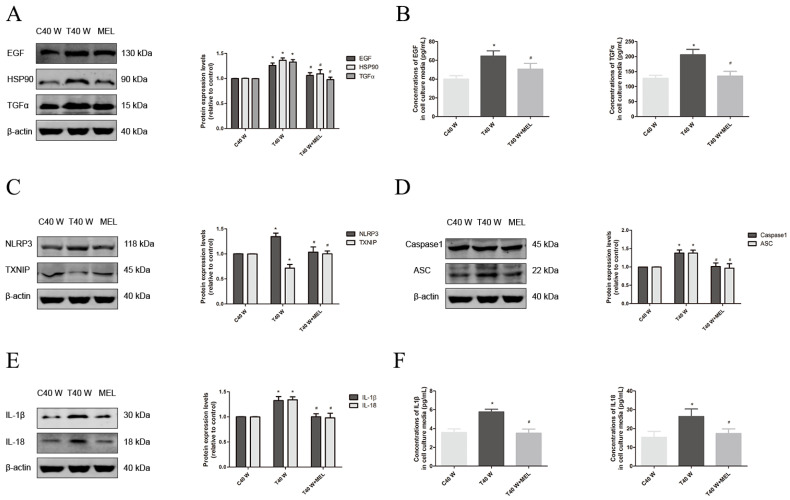
Effects of melatonin on the enhancement of HSP90, TGFα, and EGF levels and activation of linked pathways following long-term arsenite treatment. The antioxidant melatonin was added to cells treated continuously with or without 0.5 µM arsenite for 40 weeks. (**A**,**C**–**E**,**G**–**I**) Western blots showing the effects on EGF, TGFα, and HSP90 (**A**), TXNIP and NLRP3 (**C**), ASC and Caspase1 (**D**), IL-18 and IL-1β (**E**), AKT, JNK, p-AKT, and p-JNK (**G**), ERK, p38, p-ERK, and p-p38 (**H**), and p65 and p-p65 (**I**) protein levels. (**B**,**F**) Effects on EGF and TGFα (**B**), IL-18/1β (**F**) levels in cell culture media. The outcomes are reported as means ± SD, and a minimum of three separate experiments were conducted. * *p* < 0.05, vs. control group (untreated cells). # *p* < 0.05, vs. arsenite-treated group. C40 W, cells cultured for 40 weeks without arsenite treatment; T40 W, cells cultured for 40 weeks with arsenite treatment; MEL, melatonin.

**Figure 5 cells-14-00912-f005:**
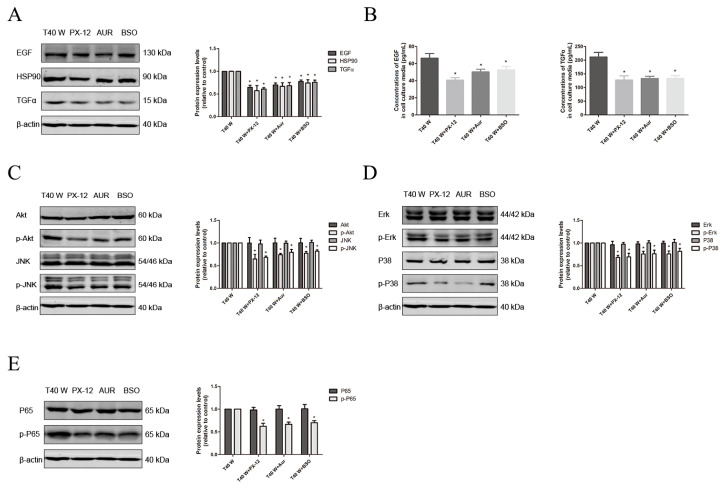
Effects of Trx1, TrxR1, and GSH on the enhancement of HSP90, TGFα, and EGF levels and activation of linked pathways following long-term arsenite treatment. Either PX-12, auranofin, or BSO was added to cells treated continuously with 0.5 µM arsenite for 40 weeks. (**A**,**C**–**E**) Western blots showing the effects on protein levels of EGF, TGFα, and HSP90 (**A**), AKT, JNK, p-AKT, and p-JNK (**C**), ERK, p38, p-ERK, and p-p38 (**D**), and p65 and p-p65 (**E**). (**B**) Effects on EGF and TGFα levels in cell culture media. The outcomes are reported as means ± SD, and a minimum of three separate experiments were conducted. * *p* < 0.05, vs. arsenite-treated group. T40 W, cells cultured for 40 weeks with arsenite treatment; Aur, auranofin.

**Figure 6 cells-14-00912-f006:**
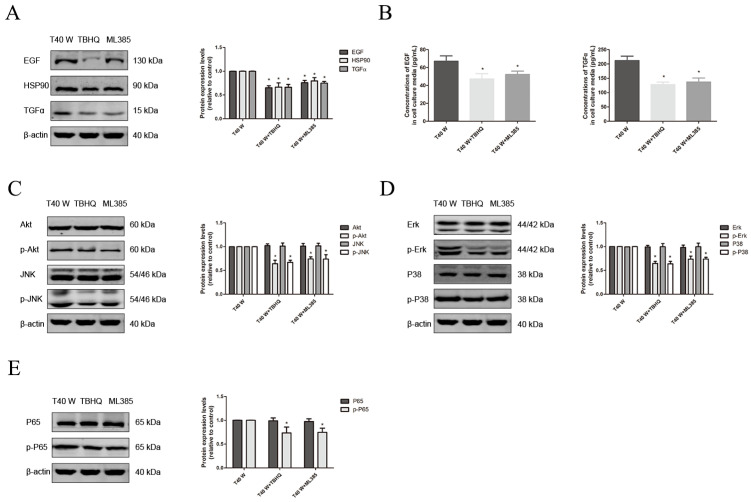
Effects of Nrf2 system on the enhancement of HSP90, TGFα, and EGF levels and activation of linked pathways following long-term arsenite treatment. Either TBHQ or ML385 was added to cells treated continuously with 0.5 µM arsenite for 40 weeks. (**A**,**C**–**E**) Western blots showing the effects on protein levels of EGF, TGFα, and HSP90 (**A**), AKT, JNK, p-AKT, and p-JNK (**C**), ERK, p38, p-ERK, and p-p38 (**D**), and p65 and p-p65 (**E**). (**B**) Effects on EGF and TGFα levels in cell culture media. The outcomes are reported as means ± SD, and a minimum of three separate experiments were conducted. * *p* < 0.05, vs. arsenite-treated group. T40 W, cells cultured for 40 weeks with arsenite treatment.

**Figure 7 cells-14-00912-f007:**
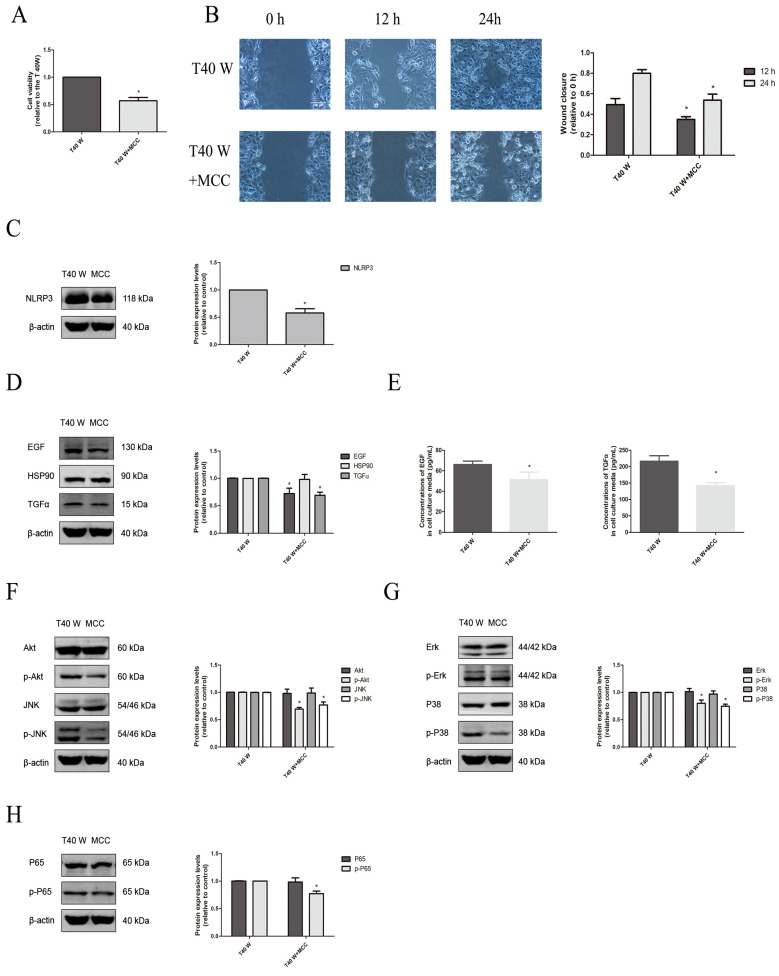
Effects of the NLRP3 inflammasome on cell proliferation and migration, enhancement of HSP90, TGFα, and EGF, and activation of related signaling pathways following long-term arsenite treatment. MCC950 was added to cells treated continuously with 0.5 µM arsenite for 40 weeks. (**A**) Cell viability was determined according to the MTS assay. (**B**) Cell migration capacities were ascertained using wound-healing assays. (**C**,**D**,**F**–**H**) Western blots showing the effects on protein levels of NLRP3 (**C**), EGF, TGFα, and HSP90 (**D**), AKT, JNK, p-AKT, and p-JNK (**F**), ERK, p38, p-ERK, and p-p38 (**G**), and p65 and p-p65 (**H**). (**E**) Effects on EGF and TGFα levels in cell culture media. The outcomes are reported as means ± SD, and a minimum of three separate experiments were conducted. * *p* < 0.05, vs. arsenite-treated group. T40 W, cells cultured for 40 weeks with arsenite treatment; MCC, MCC950. The scale bars in the photos represent 100 μM for wound healing assay.

**Figure 8 cells-14-00912-f008:**
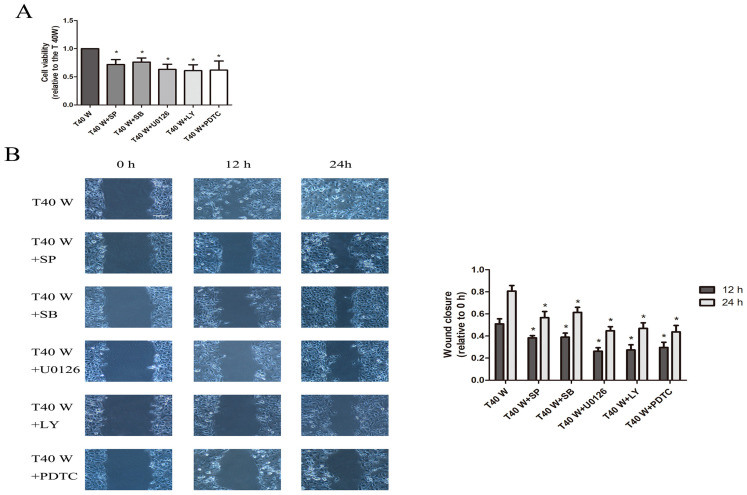
Effects of JNK, p38, ERK, AKT, or NF-κB on the cell proliferation and migration and enhancement of HSP90, TGFα, and EGF following long-term arsenite treatment. Either SP600125, SB203580, U0126, LY294002, or PDTC was added to cells treated continuously with 0.5 µM arsenite for 40 weeks. (**A**) Cell viability was determined according to the MTS assay. (**B**) Cell migration capacities were estimated using wound-healing assays. (**C**,**E**) Western blots showing the effects on protein levels of EGF, TGFα, and HSP90 (**C**), and p65 and p-p65 (**E**). (**D**) Effects on EGF and TGFα levels in cell culture media. The outcomes are reported as mean ± SD, and a minimum of three separate experiments were conducted. * *p* < 0.05, vs. arsenite-treated group. T40 W, cells cultured for 40 weeks with arsenite treatment; SP, SP600125; SB, SB203580; LY, LY294002. The scale bars in the photos represent 100 μM for wound healing assay.

## Data Availability

The datasets used and/or analyzed during the current study are available from the corresponding author on reasonable request.

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
