# Peer review of "Long-Term Treatment with Low-Level Arsenite Induces Aberrant Proliferation and Migration via Redox Rebalance in Human Urothelial Cells"

_cells, 2025, doi:10.3390/cells14120912_

Round 1
Reviewer 1 Report
Comments and Suggestions for Authors
This manuscript investigates the role of redox homeostasis in arsenite-induced aberrant proliferation and migration in human bladder uroepithelium. Using an in vitro model of chronic arsenite exposure, the authors report significant upregulation of ROS, antioxidant systems (GSH, Trx1, Nrf2), and inflammatory markers (NLRP3 inflammasome), which lead to increased cell proliferation and migration via activation of ERK, JNK, p38 MAPK, PI3K/AKT, and NF-κB pathways.
Recommendations for Improvement:
-
Materials and Methods:
-
Please include the catalogue numbers and suppliers for all antibodies, reagents, and kits used. This information is essential to ensure reproducibility.
-
Provide more detailed descriptions in the methodology section, particularly specifying concentrations, incubation periods, and exact procedures, as current descriptions are superficial in several areas, hindering reproducibility.
-
-
Quality of Figures:
-
Improve the quality and resolution of the images. The current resolution makes it difficult to analyze the figures, especially when printed or viewed without significant zoom on a computer, thus compromising detailed analysis.
-
Figures 4 and 5 contain many panels and appear overly crowded. Simplifying these figures or separating them into clearer, distinct figures would greatly enhance readability and interpretability.
-
-
Discussion:
-
Expand the discussion to explicitly include future research directions. Clearly outlining unanswered questions and potential follow-up experiments would significantly strengthen this section.
-
Provide a more critical analysis of the existing literature, particularly highlighting any contradictory findings, if available. This would deepen the scientific rigor and contextual relevance of your findings.
-
-
Writing Quality:
-
Conduct a thorough grammatical and stylistic revision of the manuscript to enhance readability and present a more professional, clear, and concise text.
-
Author Response
Reviewer #1: Comments
Materials and Methods:
- Please include the catalogue numbers and suppliers for all antibodies, reagents, and kits used. This information is essential to ensure reproducibility.
Response: According to this comment, the Materials and Methods has been rewritten based on the reviewer’s suggestion in the revised version of manuscript.
2.Provide more detailed descriptions in the methodology section, particularly specifying concentrations, incubation periods, and exact procedures, as current descriptions are superficial in several areas, hindering reproducibility.
Response: According to this comment, the Materials and Methods has been rewritten based on the reviewer’s suggestion in the revised version of manuscript.
Quality of Figures:
3.Improve the quality and resolution of the images. The current resolution makes it difficult to analyze the figures, especially when printed or viewed without significant zoom on a computer, thus compromising detailed analysis.
Response: According to this comment, all the figures will be uploaded separately after increasing resolution and we thank the reviewer very much for your carefulness.
Figures 4 and 5 contain many panels and appear overly crowded. Simplifying these figures or separating them into clearer, distinct figures would greatly enhance readability and interpretability.
Response: According to this comment, Figures 4 and 5 have been revised, and we thank the reviewer very much for your carefulness.
Discussion:
4.Expand the discussion to explicitly include future research directions. Clearly outlining unanswered questions and potential follow-up experiments would significantly strengthen this section.
Response: According to this comment, the Discussion has been rewritten based on the reviewer’s suggestion in the revised version of manuscript.
- Provide a more critical analysis of the existing literature, particularly highlighting any contradictory findings, if available. This would deepen the scientific rigor and contextual relevance of your findings.
Response: According to this comment, the Discussion has been rewritten based on the reviewer’s suggestion in the revised version of manuscript.
Writing Quality:
- Conduct a thorough grammatical and stylistic revision of the manuscript to enhance readability and present a more professional, clear, and concise text.
Response: We have already polished the article in a professional place before submitting it. If you feel that further modifications are needed, we can polish the article again.

Reviewer 2 Report
Comments and Suggestions for Authors
This paper examines the impact of long-term arsenite exposure on a bladder cancer derived cell line. The goal seems to be to relate exposure to oxidant/antioxidant balance and proliferative/migratory properties of of the cells. This approached by use of inhibitors and in one case an activator of components of pathways. One strength is use of compounds instead of genetic manipulation, so that outcomes might model attempts at therapy by chemotherapy. The singular choice of melatonin as an antioxidant is a weakness. To use it and ignore its possible pro-apoptotic effects is not good. It might be desirable to incorporate apoptosis as another parameter. The relationship between ROS levels and cell viability is not clean and ROS levels do not seem clearly related to GSH. GR, TRX, TRXR levels. One case where there might be a connection is for TXNIP, which is examined once and never shown again. Given its interaction with TRX and and other major participants in redox balance and cancer metabolism, it deserves more analysis. There is a reasonably priced TXNIP inhibitor,SRI-3730 available from venders, like Cayman, that could be used.
The Methods section should be more detailed. I suggest including such a section as a supplemental section. I don't understand the viability data as presented. Healthy cell cultures should show 80-95% live cells. How can arsenite exposure improve viability by 1.4-1.5-fold? Please detail the methods and possibly show the real and not normalized data. Was the DCFA data corrected for artifactual oxidation in cell culture media in any way? Are you linking ROS to any specific oxidant species, HOOH, lipid-OOH etc? In general, explain all details in this report and don't send readers to other publications, which may not be completely accessible. I had issues trying to get access.
Author Response
Reviewer #2: Comments
Comments and Suggestions for Authors
1.This paper examines the impact of long-term arsenite exposure on a bladder cancer derived cell line. The goal seems to be to relate exposure to oxidant/antioxidant balance and proliferative/migratory properties of of the cells. This approached by use of inhibitors and in one case an activator of components of pathways. One strength is use of compounds instead of genetic manipulation, so that outcomes might model attempts at therapy by chemotherapy. The singular choice of melatonin as an antioxidant is a weakness. To use it and ignore its possible pro-apoptotic effects is not good. It might be desirable to incorporate apoptosis as another parameter. The relationship between ROS levels and cell viability is not clean and ROS levels do not seem clearly related to GSH. GR, TRX, TRXR levels. One case where there might be a connection is for TXNIP, which is examined once and never shown again. Given its interaction with TRX and and other major participants in redox balance and cancer metabolism, it deserves more analysis. There is a reasonably priced TXNIP inhibitor,SRI-3730 available from venders, like Cayman, that could be used.
Response: Thank you for your suggestion. We will listen to your suggestions for our upcoming experiments and look forward to publishing future data in Cells.
- The Methods section should be more detailed. I suggest including such a section as a supplemental section. I don't understand the viability data as presented. Healthy cell cultures should show 80-95% live cells. How can arsenite exposure improve viability by 1.4-1.5-fold? Please detail the methods and possibly show the real and not normalized data. Was the DCFA data corrected for artifactual oxidation in cell culture media in any way? Are you linking ROS to any specific oxidant species, HOOH, lipid-OOH etc? In general, explain all details in this report and don't send readers to other publications, which may not be completely accessible. I had issues trying to get access.
Response: According to this comment, the Materials and Methods has been rewritten based on the reviewer’s suggestion in the revised version of manuscript and we thank the reviewer very much for your carefulness.
We have tried our best to revise our manuscript according to these suggestions, and hope that the new revision will meet with approval. However, if any further information is still required or some parts need to be rewritten, please don’t hesitate to contact us at your convenient time.
Looking forward to hearing from you.
Best regards.
Peiyu Jin
China Medical University

Round 2
Reviewer 1 Report
Comments and Suggestions for Authors
The manuscript would benefit from substantial English editing to improve clarity and readability.
Author Response
Reviewer #1: Comments
1.The manuscript would benefit from substantial English editing to improve clarity and readability.
Response: We have already conducted substantial English editing in MDPI Author Services. English Editing ID: english-95239.

Reviewer 2 Report
Comments and Suggestions for Authors
While improved, the paper still does not address all my concerns. Please add a line about the impact of long-term arsenic exposure on cell viability, must be fairly strong to show increases attributed to treatments. Add a line in the discussion about your promise to investigate TXNIP further.
Author Response
Reviewer #2: Comments
1.While improved, the paper still does not address all my concerns. Please add a line about the impact of long-term arsenic exposure on cell viability, must be fairly strong to show increases attributed to treatments.
Response: According to this comment, we have added this sentence in the results and highlighted it in yellow, and we thank the reviewer very much for your carefulness.
In the uroepithelium, long-term exposure to low-dose arsenite may induce cell self-renewal, apoptosis resistance, energy metabolism reprogramming, enhanced glucose uptake and aerobic glycolysis, which contributes to increases in cell viability and proliferation.
2.Add a line in the discussion about your promise to investigate TXNIP further
Response: According to this comment, we have added this sentence in the discussion and highlighted it in yellow, and we thank the reviewer very much for your carefulness.
We plan to further investigate the mechanism of TXNIP changes in the uroepithelium when chronically treated with arsenite.
We have tried our best to revise our manuscript according to these suggestions, and hope that the new revision will meet with approval. However, if any further information is still required or some parts need to be rewritten, please don’t hesitate to contact us at your convenient time.
Looking forward to hearing from you.
Best regards.
Peiyu Jin
China Medical University

Round 3
Reviewer 2 Report
Comments and Suggestions for Authors
OK
Author Response
We have tried our best to revise our manuscript according to these suggestions, and hope that the new revision will meet with approval. However, if any further information is still required or some parts need to be rewritten, please don’t hesitate to contact us at your convenient time.
Looking forward to hearing from you.
Best regards.
Peiyu Jin
China Medical University